# Assessment of Metabolic Parameters in Female Triathletes with Hashimoto’s Thyroiditis in Poland

**DOI:** 10.3390/biomedicines11030769

**Published:** 2023-03-02

**Authors:** Marcin Gierach, Roman Junik

**Affiliations:** 1Department of Endocrinology and Diabetology, Collegium Medicum in Bydgoszcz, Nicolaus Copernicus University in Toruń, ul. M. Skłodowskiej-Curie 9, 85-094 Bydgoszcz, Poland; 2Cardiometabolic Center Gierach-Med, ul. Bydgoskich Olimpijczyków 5/39-40, 85-796 Bydgoszcz, Poland

**Keywords:** body mass index, Hashimoto’s thyroiditis, female triathletes, fat mass, skeletal muscle mass

## Abstract

**Background**: Hypothyroidism is a complex disorder characterized by an increase in body weight. About 15–30% of hypothyroid patients are reported to be overweight. The triathlon is an endurance combination sport that comprises a sequential swim, cycle, and run. Triathletes must withstand high training loads with various combinations of intensity and volume. Adequate body structure, the ratio of fat and muscle tissue, and adequate hydration play a huge role. The aim of our study was to show the potential differences in metabolic parameters assessed by medical Body Composition Analyzer before the initiation of treatment with L-thyroxine and after 3 and 6 months of treatment in females who practiced triathlon and who were newly diagnosed with Hashimoto’s thyroiditis. **Methods**: The study group included 32 females practicing triathlon. They were recruited for 10 months from March to December 2021. Analysis of anthropometric measurements was performed using a seca device mBCA 515 medical Body Composition Analyzer. **Results**: We observed significant differences in FM and VAT before and after L-thyroxine treatment. We also noticed lower BMI levels after treatment, along with significant differences in thyroid function tests (TSH and fT4) carried out during the recruitment period and after 3 and 6 months of treatment. Conclusion: Due to their higher daily energy consumption, further research is needed into the treatment of Hashimoto’s thyroiditis in athletes who practice triathlon. Frequent bioelectrical impedance analysis of body composition during treatment can be very helpful.

## 1. Introduction

In Poland, up to 22% of the population may have thyroid problems. This mainly applies to women [1]. Hypothyroidism is one of the most common thyroid disorders and is often caused by chronic autoimmune inflammation—affecting approximately 4–10% of the population—characterized by the production of anti-thyroid peroxidase antibodies (TPOAb) and anti-thyroglobulin antibodies (TgAb) [2]. Hashimoto’s thyroiditis (HT) is a polygenic disease with still not fully defined etiopathogenesis. In Poland, we observe a constant increase in the incidence of this disease in the female population of all ages, but especially in young women of childbearing age. Women are about eight times more likely to develop HT than men.

Thyroid dysfunction (hypothyroidism) not only alters the appetite and makes it difficult to maintain normal body weight, but it also affects body composition, regardless of physical activity.

Thyroid hormones (triiodothyronine-T3, thyroxine-T4) are considered to be strong modulators of thermogenesis, and their deficiency predisposes to the development of central obesity [3,4,5]. Even subclinical hypothyroidism may affect weight gain, so it can be a risk factor for overweight and obesity.

Many studies concerning the association between thyroid dysfunction and changes in body weight are based on the analysis of elementary indicators, for example, body weight or body mass index (BMI) [6,7,8,9]. There is a need for a more thorough analysis of these parameters, to enable information to be obtained on the content of muscle tissue, water, and fat in the body.

Triathlon is a combined endurance sport that involves sequential swimming, transition from swimming to cycling (T1), cycling, transition from cycling to running (T2), and running various “long” or “short” distances [10]. Triathletes have to endure high training loads with various combinations of intensity and volume [11]. Adequate body structure, the ratio of fat and muscle tissue, and adequate hydration play a huge role.

The aim of our study was to show the potential differences in the metabolic parameters assessed by medical Body Composition Analyzer (mBCA) before the initiation of treatment with L-thyroxine and after 3 and 6 months of treatment in females with newly diagnosed Hashimoto’s thyroiditis (HT), who were practicing triathlon.

## 2. Material and Methods

### 2.1. Participants

The study group included 32 females practicing triathlon. They were recruited for 10 months from March to December 2021 by the Cardiometabolic Center Gierach-Med in Bydgoszcz, Poland and the Department of Endocrinology and Diabetology Collegium Medicum University of Nicolaus Copernicus in Bydgoszcz, Poland. All the patients provided verbal consent to participate in the study. The mean age of the women studied was 33 years, and standard deviation was ±4.76 years. The average height of the triathletes was 169.6 ± 4.7 cm, body weight was 68.05 ± 4.83 kg, and BMI was 23.78 ± 1.02 kg/m^2^. The average number of hours a week spent on training by the participants was 8.71 ± 1.26 h.

Exclusion criteria for the study were patients using drugs which could affect thyroid functions, such as lithium, amiodarone, steroids, beta blockers, or interferon; patients using drugs which could affect body water–lipid homeostasis, such as diuretics or oral contraceptives; smoking, chronic renal failure, hepatic failure, congestive heart diseases, malnutrition, malignant diseases, pregnant women, and patients with other known endocrine disorders.

### 2.2. Body Composition Analysis

Analysis of the anthropometric measurements was performed using a seca device mBCA 515 medical Body Composition Analyzer. The measurements were taken three times (during recruitment, then at 3 and 6 months after initiation of L-thyroxine treatment) in the morning on an empty stomach. There was no training the day before the study. Participants were also informed not to consume alcohol 24 h before the measurements, and not to consume caffeine, including beverages, 4 h before the measurements, in accordance with the manufacturer’s instructions. All participants wore light clothing, and earrings, rings, bracelets, and any metal which could influence the results, were removed before the measurements were taken.

The device measures the composition of the body by bioelectrical impedance analysis (BIA), using a pair of electrodes for each hand and foot. The performance and accuracy of every BIA device depends directly on its validation and reference data, and seca mBCA passed an extensive scientific validation process [12,13,14,15]. We assessed the following parameters: body mass index (BMI); fat mass (FM); fat-free mass (FFM); skeletal muscle mass (SMM); total body water (TBW); extracellular water (ECW); resistance (BIWA); visceral adipose tissue (VAT).

Patients were then started on L-thyroxine supplements. The dose of the drug was increased periodically (every 4–6 weeks) stepwise, based on TSH estimations, until the patients were rendered euthyroid, that is, had a TSH level between 0.5 and 4.5 µIU/l. The mean dose during the study was 65.5 µg per day with a standard deviation (SD) ± 12.3 µg. The subjects were advised not to change their dietary and exercise habits. The characteristics of the study group are presented in Table 1.

### 2.3. Thyroid Gland Ultrasonography

The measurement of the thyroid gland was performed using an ultrasound scan (US) with a 10 MHz linear probe using the Vivid S60N. The US was performed in a darkened room in a lying position with the head tilted back. The structure of the thyroid gland and its dimensions were assessed by one endocrinologist.

### 2.4. Biochemical Analyses

Venous blood samples were collected from fasting patients for biochemical analyses (TSH, fT3, fT4, TPOAbs, TgAbs). The diagnosis of Hashimoto’s thyroiditis (HT) was based on the presence of a hypoechogenic thyroid structure on US examination and elevated serum concentration of thyroid peroxidase antibodies (TPOAbs) and/or antibodies against thyroglobulin (TgAbs) [16]. HT was diagnosed on the basis of typical clinical symptoms (tiredness, weakness, dry skin, feeling cold, etc.), decreased fT3 and/or fT4 and decreased TSH and, additionally, the presence of anti-thyroid antibodies. All the tests were performed at the Department of Laboratory Medicine, Nicolaus Copernicus University, Collegium Medicum, Bydgoszcz, Poland using a Horiba ABX Pentra 400 analyzer (Horiba ABX, Montpelier, France).

### 2.5. Statistical Analyses

Statistical analysis was performed using the Statistica 10.0 software (Statsoft, Bydgoszcz, Poland). The results were expressed as mean of ± standard deviation (SD). The Kruskal–Wallis test for independent variables was used for the comparison of the groups, followed by the ANOVA test. The results were considered statistically significant when *p* < 0.01.

### 2.6. Ethic Approval

All the procedures used in the present study were performed in accordance with the 1964 Helsinki declaration and its later amendments and other relevant guidelines and regulations. The research protocol was reviewed and approved by the Ethics Committee at the University Hospital in Bydgoszcz (Permission number KB/224/2022). All subjects granted their informed consent for participation in the study.

## 3. Results

We observed significant differences in FM and VAT before and after L-thyroxine treatment. We also noticed lower BMI levels after treatment (23.78 vs. 23.22 vs. 23.14, respectively), and higher levels of SMM (21.2 vs. 21.5 vs. 21.6, respectively), but there were no significant differences. We did not find any differences in FFM or BIVA (Table 2).

We also observed significant differences in thyroid function tests (TSH and fT4) carried out during the recruitment period and after 3 and 6 months of treatment (TSH: 4.84 vs. 2.31 vs. 1.93, respectively, and fT4: 9.26 vs. 11.23 vs. 12.87, respectively). There were no significant differences related to TPOAbs and TgAbs (Table 3).

The results of correlation analysis between serum thyroid hormone levels (TSH, fT4) and metabolic parameters in participants are shown in Table 4.

We found no obvious associations between serum thyroid hormone levels (TSH, fT4) and FM, FFM, SSM, and TBW. There was a significant negative correlation of serum FT4 levels with BMI and VAT among participants with HT (*r* = −0.087, *p* < 0.01; and *r* = −0.125, *p* < 0.01, respectively). Serum TSH levels were also positively correlated with BMI and VAT (*r* = 0.236, *p* < 0.01; and *r* = 0.324, *p* < 0.01, respectively) (Table 4).

After 3 months of treatment with L-thyroxine we also found that serum FT4 levels showed a negative correlation with BMI and VAT (*r* = −0.065, *p* < 0.01; and *r* = −0.088, *p* < 0.01, respectively), and serum TSH levels showed a positive correlation with BMI (*r* = 0.202, *p* < 0.01) and also a positive correlation with VAT (*r* = 0.277, *p* < 0.01) in the participants with HT. Similarly, there were no significant associations between FFM, FM, SSM, and TBW (Table 5).

After 6 months of treatment with L-thyroxine we observed an association between TSH levels and BMI and VAT. There was a positive correlation (*r* = 0.255, *p* < 0.01; and *r* = 0.223, *p* < 0.01, respectively). We also found that serum FT4 levels showed a negative correlation with BMI and VAT (*r* = −0.125, *p* < 0.01; and *r* = −0.123, *p* < 0.01, respectively). Similarly, there were no significant associations between FFM, FM, SSM, and TBW (Table 6).

## 4. Discussion

Hypothyroidism is a complex disorder characterized by an increase in body weight. About 15–30% of hypothyroid patients are reported to be overweight. In their study, Malczyk et al. [17] observed that women with HT were characterized by significantly higher values of body weight, and thus BMI index, than healthy women (73.64 kg vs. 64.36 kg, *p* < 0.0001; 27.65 kg/m^2^ vs. 23.95 kg/m^2^, *p* < 0.001, respectively). In our group, the initial BMI in triathletes was normal (23.78 ± 1.02 kg/m^2^), which is probably due to their previous high level of physical activity (8.71 ± 1.26 h per week), and it decreased, although statistically insignificantly, during L-thyroxine treatment. However, Okan et al. did not observe a decrease in body weight and body fat percentages, in spite of the fact that the participants achieved euthyroidism with adequate L-thyroxine replacement [18]. In other studies, obesity also seems to persist, even after established euthyroidism via levothyroxine (LT4) replacement therapy [19,20].

Hypothyroidism is known to lead to an increase in weight and fat content in the body. Sanyala and Raychaudhuri, in their study, noticed that hormonal disorders connected with Hashimoto’s disease affected not only the change in body weight, but also its composition, regardless of physical activity [5]. As reported by Malczyk et al., the problem of body fat excess affected women with HT more often than healthy women (44.7% vs. 13.8%, *p* < 0.001) [17], and accounted for the decrease in thermogenesis, fat tissue metabolism, and fluid retention [3]. Adipose tissue is an active internal endocrine organ, but it is also responsible for energy storage. It consists of different types of cells, such as fat cells, fibroblasts, and immune cells, and it is divided into two types—brown and white. Brown adipose tissue (BAT) plays a key role in the process of thermogenesis and also in maintaining normal body weight. It has been observed that the amount of BAT decreases with increasing body mass. Thus, the right amount of active BAT can prevent obesity in adults [21]. Adipose tissue, as an endocrine organ, secretes various biologically active substances [22,23], one of which is leptin, which can play an important role in the interaction between the composition of the body and thyroid hormones [24]. There are studies which show that leptin may stimulate TSH secretion and influence peripheral conversion of T4 to T3, and it may be associated with thyroid gland autoimmunity [25,26].

The bioelectrical impedance analysis system (BIA) is an easy and a cheap method which helps to identify different body compartments: body lipid percentage, fat-free body mass, and total body fluid [27]. In our study we used a seca device mBCA 515 medical Body Composition Analyzer for analysis of anthropometric measurements. We observed significant differences in FM and VAT before and after 6 months of treatment with L-thyroxine. In hypothyroid patients, body fat content is shown to increase in parallel with body weight. This effect is attributed to the reduction of lipid metabolism in hypothyroidism [28,29]. In their study, Malczyk et al., observed that the problem of excess of fat in the body affected women with HT more often than healthy women (44.7% vs. 13.8%, *p* < 0.001) [18]. Nevertheless, in a few other studies, no significant reduction in body fat was observed with L-thyroxine therapy [30,31]. Ruhla et al. also reported that L-thyroxine therapy is associated with an increase in BMI independent from the level of TSH [20]. Karmisholt et al. observed that weight loss in hypothyroid subjects treated with L-thyroxine resulted mainly from the excretion of excess body water associated with myxedema, and not from a change in adipose tissue [32].

There was a significantly positive correlation of serum TSH levels with BMI and VAT among participants with HT (*r* = 0.236, *p* < 0.01; and *r* = 0.324, *p* < 0.01, respectively) in our study. Many authors have attributed this relationship to the adipokine leptin. Leptin regulates thyrotropin-releasing hormone (TRH) gene expression in the paraventricular nucleus, and TSH, in turn, increases leptin production from adipose tissue. Leptin also regulates the conversion of T4 to T3 [4,33]. The strongest correlation was established between TSH and BMI (*r* = 0.23). These findings are compatible with some previous studies [34,35,36,37]. In a study by Yasar et al., which correlated thyroid function with obesity in a cohort of polycystic ovary syndrome subjects, the correlation of TSH with BMI was also significant (*r*: 0.122; *p*: 0.02) [36]. A review by Amanda de Moura Souza published in 2011 analyzed data from 29 studies. Some of these studies demonstrated a positive correlation between body mass index and TSH, but approximately half of the studies showed no such correlation [7]. The hypothesis for correlation was that TSH is involved in the differentiation of pre-adipocytes and induced adipogenesis. Another hypothesis is the leptin hypothesis. Some studies have demonstrated a positive correlation between leptin and TSH [33].

In our study, there was also a significantly negative correlation of serum FT4 levels with BMI and VAT among participants with HT (*r* = −0.087, *p* < 0.01; and *r* = −0.125, *p* < 0.01, respectively). Kim B et al. reported that FT4 was positively associated with blood pressure, FPG, HDL, and triglyceride levels, and negatively associated with waist circumference in euthyroid subjects [38]. Pratz-Puig et al. also demonstrated that a free thyroxine level (FT4) close to the lower limit is related to increased body mass index, visceral fat, and insulin resistance [39].

## 5. Limitation of Our Study

The main limitation of the study was the small numbers recruited and the limited duration of follow-up. Diet, supplementation, and modifications in training used by triathletes were not taken into account. Throughout the study, patients were advised not to change their dietary and exercise habits.

## 6. Conclusions

The problem of treating Hashimoto’s thyroiditis in athletes who practice triathlon requires further research due to the higher daily energy consumption caused by heavy physical exertion, adequate supplementation, and diet. Frequent bioelectrical impedance analysis of body composition analysis during treatment can be very helpful.

## Figures and Tables

**Table 1 biomedicines-11-00769-t001:** The general characteristics of the study group of 32 females practicing triathlon.

Parameters	Recruitment Period ± SD	After 6 Months ± SD	*p* < 0.01
Tch [mg/dL] N < 190	187 ± 12	178 ± 11	NS
LDL-C [mg/dL] N < 115	103 ± 11	98 ± 10	NS
HDL-C [mg/dL] N > 45	53 ± 8	56 ± 7	NS
Non-HDL-C [mg/dL] N < 145	132 ± 14	126 ± 13	NS
TG [mg/dL] N < 150	124 ± 16	119 ± 15	NS
Glucose [mg/dL] N-70-99	88 ± 6	86 ± 6	NS
Creatinine [mg/dL] N-0.6-1.3	0.93 ± 0.09	0.89 ± 0.08	NS
ALT [U/L] N-5-40	32 ± 4	31 ± 4	NS
Calcium [mg/dL] N-2.25-2.65	2.31 ± 0.12	2.35 ± 0.13	NS
Iron [µg/dL] N-60-180	89 ± 16	93 ± 18	NS
Magnesium [mmol/L] N-0.65-1.2	0.87 ± 0.12	0.89 ± 0.11	NS

Tch: total cholesterol; LDL-C: low density lipoprotein cholesterol; HDL-C: high density lipoprotein cholesterol; Non-HDL-C: non-high density lipoprotein cholesterol; TG: triglycerides; ALT: alanine aminotransferase.

**Table 2 biomedicines-11-00769-t002:** Assessment of metabolic parameters in triathletes before L-thyroxine treatment and 3 and 6 months after starting treatment.

Parameters	Recruitment Period	3 Months	6 Months	*p* < 0.01
BMI [kg/m^2^]	23.78 ± 4.57	23.22 ± 4.33	23.14 ± 4.16	NS
FM [kg/m^2^]	21.16 ± 1.8	20.32 ± 1.6	19.27 ± 1.5	<0.01
**FM [%]**	31.1%	30.2%	29.2%	<0.01
FFM [kg/m^2^]	46.88 ± 2.45	46.64 ± 2.23	46.23 ± 2.27	NS
FFM [%]	68.9%	69.2%	70.8%	NS
SMM [kg]	21.23 ± 1.23	21.56 ± 1.12	21.62 ± 1.13	NS
SMM [%]	31.1%	31.6%	31.5%	NS
TBW [l/%]	29.7/43.6	30.2/44.5	30.5/44.7	NS
ECW [l/%]	12.6/18.5	13.2/18.9	13.4/19.2	NS
ECW/TBW [%]	42.42	43.71	43.93	NS
BIWA [Ω]	643.9	622.4	613.3	NS
**VAT [l]**	0.8	0.6	0.4	<0.01

BMI: body mass index; FM: fat mass; FFM: fat-free mass; SMM: skeletal muscle mass; TBW: total body water; ECW: extracellular water; BIWA: resistance; VAT: visceral adipose tissue.

**Table 3 biomedicines-11-00769-t003:** Comparison of thyroid function tests before and 3 and 6 months after starting L-thyroxine treatment. The mean dose during the study was 65.5 µg per day with a standard deviation (SD) ± 12.3 µg.

Thyroid Parameters	Recruitment Period	3 Months	6 Months	*p* < 0.01
TSH [uIU/mL] (N-0.35-4.55) + SD	4.84 ± 1.44	2.31 ± 1.12	1.93 ± 0.87	<0.01
fT4 [pmol/L] (N-7.3-14.4) + SD	9.26 ± 2.15	11.23 ± 2.54	12.87 ± 2.66	<0.01
TPOAbs [IU/mL] (N-0-9) + SD	289 ± 76	266 ± 74	263 ± 66	NS
TgAbs [IU/mL] (N-0-4) + SD	192 ± 57	186 ± 55	172 ± 62	NS
L-thyroxine- average dose + SD	-	58.4 ± 10.2	73.2 ± 14.4	NS

TSH: thyroid-stimulating hormone; fT3: free triiodothyronine; fT4: free thyroxine; TPOAbs: thyroid peroxidase antibodies; TgAbs: antibodies against thyroglobulin; SD: standard deviation.

**Table 4 biomedicines-11-00769-t004:** Correlation between thyroid function tests (TSH, fT4) and metabolic parameters-before treatment.

Metabolic Parameters	TSH		fT4		TPOAbs		TgAbs	
	*r*	*p*	*r*	*p*	*r*	*p*	*r*	*p*
BMI	0.236	<0.01	−0.087	<0.01	0.113	NS	0.094	NS
VAT	0.324	<0.01	−0.125	<0.01	0.142	NS	0.155	NS
FM	0.126	NS	−0.056	NS	0.008	NS	0.006	NS
FFM	0.142	NS	0.004	NS	0.013	NS	0.021	NS
SSM	0.015	NS	0.012	NS	0.043	NS	0.023	NS
TBW	−0.047	NS	0.013	NS	0.003	NS	0.008	NS

Data are derived from Spearman correlation coefficient. The level of significance was accepted at *p* < 0.01. BMI: body mass index; VAT: visceral adipose tissue; FM: fat mass; FFM: fat-free mass; SMM: skeletal muscle mass; TBW: total body water; TSH: thyroid-stimulating hormone; fT4: free thyroxine; TPOAbs: thyroid peroxidase antibodies; TgAbs: antibodies against thyroglobulin.

**Table 5 biomedicines-11-00769-t005:** Correlation between thyroid function tests (TSH, fT4) and metabolic parameters—3 months after treatment.

Metabolic Parameters	TSH		fT4		TPOAbs		TgAbs	
	*r*	*p*	*r*	*p*	*r*	*p*	*r*	*p*
BMI	0.202	<0.01	−0.065	<0.01	0.084	NS	0.042	NS
VAT	0.277	<0.01	−0.088	<0.01	0.075	NS	0.038	NS
FFM	0.112	NS	−0.072	NS	0.021	NS	0.032	NS
FM	0.089	NS	0.008	NS	0.006	NS	0.004	NS
SSM	0.023	NS	0.031	NS	0.004	NS	0.012	NS
TBW	−0.012	NS	0.009	NS	0.002	NS	0.003	NS

Data are derived from Spearman correlation coefficient. The level of significance was accepted at *p* < 0.01. BMI: body mass index; VAT: visceral adipose tissue; FM: fat mass; FFM: fat-free mass; SMM: skeletal muscle mass; TBW: total body water; TSH: thyroid-stimulating hormone; fT4: free thyroxine; TPOAbs: thyroid peroxidase antibodies; TgAbs: antibodies against thyroglobulin.

**Table 6 biomedicines-11-00769-t006:** Correlation between thyroid function tests (TSH, fT4) and metabolic parameters—6 months treatment.

Metabolic Parameters	TSH		fT4		TPOAbs		TgAbs	
	*r*	*p*	*r*	*p*	*r*	*p*	*r*	*p*
BMI	0.255	<0.01	−0.125	<0.01	0.033	NS	0.029	NS
VAT	0.223	<0.01	−0.123	<0.01	0.048	NS	0.019	NS
FFM	0.135	NS	−0.086	NS	0.044	NS	0.036	NS
FM	0.076	NS	0.013	NS	0.012	NS	0.023	NS
SSM	0.009	NS	0.008	NS	0.006	NS	0.007	NS
TBW	−0.004	NS	0.012	NS	0.004	NS	0.002	NS

Data are derived from Spearman correlation coefficient. The level of significance was accepted at *p* < 0.01. BMI: body mass index; VAT: visceral adipose tissue; FM: fat mass; FFM: fat-free mass; SMM: skeletal muscle mass; TBW: total body water; TSH: thyroid-stimulating hormone; fT4: free thyroxine; TPOAbs: thyroid peroxidase antibodies; TgAbs: antibodies against thyroglobulin.

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
