# Peer review of "Assessment of Metabolic Parameters in Female Triathletes with Hashimoto’s Thyroiditis in Poland"

_biomedicines, 2023, doi:10.3390/biomedicines11030769_

Round 1
Reviewer 1 Report
The aim of our study was to show the potential differences in metabolic parameters assessed by medical Body Composition Analyzer before the initiation of treatment with L-thyroxine and after 3- and 6-months treatment in females with newly diagnosed Hashimoto’s thyroiditis, whose practicing triathlon.
The study group included 32 females practicing triathlon.They were recruited for 10 months from March to December 2021. The analysis of the anthropometric measurements was performed using a seca device mBCA 515 medical Body Composition Analyzer.
We observed significant differences in FM and VAT before and after L-thyroxine treatment. We also noticed lower level of BMI after treatment and observed significant differences in thyroid function tests (TSH and fT4) in recruitment period and after 3- and 6-months treatment.
Conclusion: The problem of treating Hashimoto’s thyroiditis in athletes who practise triathlon requires further research due to the higher daily energy consumption. Frequent body composition analysis by Bioelectrical Impedance Analysis during treatment can be very helpful.

Author Response
Dear Reviewer,
Thank You very much for Your valuable comments. We have corrected our study according to Your recommendations.
- We added the word "females" to the title.
- We added the word "females" to key words
Yours faithfully
Marcin Gierach
Reviewer 2 Report
The study by Gierach and Junik shows the differences in metabolic parameters assessed by medical Body Composition Analyzer before and after initiation of treatment with L-thyroxine in females practicing triathlon.
The manuscript is well-written, presented in an intelligible fashion and the language is clear and correct. All procedures have been clearly described and statistical methods used are well chosen. References are up to date and appropriate.
My only concern is the size of the study group, however, taking into account its specificity (female thriatlonists with Hashimoto’s disease) the results of the study can make an important contribution to the knowledge about L-thyroxine treatment in athletes.
In my opinion, this manuscript is suitable for publication in Biomedicines.
Author Response
Dear Reviewer,
Thank You very much for Your valuable comments. We have corrected our study according to Your recommendations.
-
The size of the study group - female thriatlonists with Hashimoto's disease. In Poland more and more women practice triathlon, but it is very difficult to gather a larger group with thyroid problems. I do triathlon myself, so I know these athletes well, but despite it, I managed to take only this group. But we will try to conduct research and collect a larger group in the future.
Yours faithfully
Marcin Gierach